# Prognostic impact of misdiagnosis of cardiac channelopathies as epilepsy

**Javier Ramos-Maqueda**[1☯], **Francisco Bermúdez-Jiménez**[1,2☯], **Rosa Macías Ruiz**[1,2], **Mercedes Cabrera Ramos**[1], **Manuel Molina Lerma**[1,2], **Pablo Sánchez Millán**[1,2], **Miguel Álvarez López**[1,2], **Luis Tercedor Sánchez**[1,2], **Juan Jiménez-Jáimez**[1,2]*

1 Cardiology Department, Hospital Universitario Virgen de las Nieves, Granada, Spain, 2 Instituto de Investigación Biosanitaria de Granada, Granada, Spain

☯ These authors contributed equally to this work.
* Jimenez.jaimez@gmail.com

**Data Availability Statement:** The minimal anonymized data set is uploaded as a Supporting Information file.

## Abstract

### Introduction

Cardiac channelopathies are a frequent cause of sudden cardiac death (SCD) and often manifest with convulsive syncope, leading to a misdiagnosis of epilepsy. We aim to evaluate the clinical impact of epilepsy misdiagnosis in a cohort of patients with cardiac channelopathies.

### Methods

Fifty probands/families with a cardiac channelopathy were included. We retrospectively collected information from medical records to identify all patients who presented with convulsive syncope and were diagnosed with epilepsy after neurological evaluation. Clinical data and outcome were compared with those of patients without a previous epilepsy diagnosis.

### Results

Eight patients had a previous diagnosis of epilepsy. At first episode, 3 of them presented a positive family history of SCD and 5 showed a pathological electrocardiogram; half presented with sudden cardiac arrest (SCA) and the rest with recurrent syncope despite treatment with 1 or more anti-epileptic drugs. Five patients had long QT syndrome, 2 had catecholaminergic polymorphic ventricular tachycardia, and 1 had Brugada syndrome. Epilepsy misdiagnosis was associated with an increased risk of SCA/SCD (OR 6.92, $P$ = .04), a delay of 12 years ($P$ = .047) in correct diagnosis, and a delay from first symptom to channelopathy diagnosis of 18.45 years ($P$ < .0001).

### Conclusion

Cardiac channelopathy patients can be misdiagnosed with epilepsy. This involves a delayed diagnosis, a delay from the first symptom to a correct diagnosis, and an increased risk of SCA/SCD.

**Funding:** The author(s) received no specific funding for this work.

**Competing interests:** The authors have declared that no competing interests exist.

## Introduction

The causes of sudden cardiac death (SCD) among pediatric and young individuals (under 35 years of age) are predominantly primary arrhythmia syndromes (channelopathies) and cardio-myopathies.[1] Sudden cardiac death from channelopathies is estimated to account for 10–15% of SCD in individuals without structural heart disease at autopsy, often called sudden arrhythmic death syndrome (SADS).[2] Channelopathies are caused by mutations in genes encoding cardiac ion channel subunits or proteins that interact with, or regulate, ion channels. These genetic variants result in gain or loss of channel function, modifying ventricular action potential generation and leading to life-threatening arrhythmias. Long QT syndrome (LQTS), Brugada syndrome (BS), catecholaminergic polymorphic ventricular tachycardia (CPVT), and short QT syndrome (SQTS) are the most common channelopathies. These genetic conditions are often identified by specific ECG abnormalities either at baseline or in particular circumstances, such as during exercise (e.g., CPVT and LQTS), fever (e.g., BS), or pharmacological challenge.[3,4]

Warning symptoms may precede the SCD episode, one of the most common events being syncope.[5] Cardiogenic syncope is secondary to a rapid self-terminating polymorphic ventricular tachycardia, frequently accompanied by epileptiform activity (myoclonic seizures). These episodes are recurrent and are sometimes misdiagnosed as seizure disorders.[6] Up to 20% of patients with convulsive syncope could be misdiagnosed with an epileptic disorder.[7] As a result, patients at risk of SCD suffer a delay in diagnosis and are commonly exposed to anti-epileptic drugs (AED) with potential pro-arrhythmogenic effects.

We aim to evaluate the prevalence and clinical impact of epilepsy misdiagnosis in a well-characterized, single-center cohort of channelopathy patients.

## Methods

### Population and study groups

All subjects signed the written informed consent form. The local Ethics Committee approved the study (Virgen de las Nieves Hospital Ethical Committee. All index cases from 50 families screened for suspicion of channelopathy or during SCD evaluation at our Inherited Cardiac Disease Unit between 2012 and 2018 were included. Probands were diagnosed according to current international criteria[8] and classified in two groups according to the presence or absence of a previous diagnosis of epilepsy. A positive epilepsy diagnosis was defined as the presence of either a personal history of seizures, epilepsy or a history of AED therapy after a neurologist assessment. We assessed the presence of a diagnosis of epilepsy or seizure-related disorder in medical records from all included patients. We collected clinical information, neuroimaging and electroencephalogram (EEG) findings, and details of treatment for those individuals. Exclusion criteria included all acquired causes of seizures comprising traumatic/vascular injury, fever or metabolism disbalance.

### Clinical, genetic, and ECG variables

Individual clinical information was collected retrospectively, including personal history (especially regarding prior neurological evaluations), positive family history of SCD, symptomatology and triggers at first manifestation, 12-lead electrocardiogram (ECG), bidimensional echocardiography, exercise test and genetic evaluation according to phenotype, clinical presentation, and physician criteria. Selected patients underwent pharmacological challenge. Some patients presenting with idiopathic ventricular fibrillation (VF) were diagnosed based on a previous published protocol.[9] Clinical information on arrhythmic events was collected

for all the participants. We analyzed time and age from first event to SCA/SCD episode, as well as age at diagnosis.

First available ECGs were retrospectively collected. Baseline ECGs from probands were performed at 25 mm/s and blind-reviewed by two clinical cardiac electrophysiologists. A QT interval was considered abnormally prolonged when QTc was greater than 460 ms in females and 440 in males, paying special attention to T-wave amplitude and shape; BS was diagnosed when type 1 Brugada pattern appeared spontaneously or after ajmaline or flecainide challenge. A diagnosis of CPVT required at least three premature ventricular complexes of different morphologies during exercise test or epinephrine challenge. Short QT syndrome was diagnosed according to current criteria.[8,10]

Sudden cardiac arrest was defined as unexpected circulatory arrest reversed by successful resuscitation maneuvers. The definition of SCD applied when no obvious extra-cardiac causes had been identified by post-mortem examination and therefore an arrhythmic event was a likely cause of death.[11] Sudden unexpected death in epilepsy (SUDEP) was defined as a "sudden, unexpected, witnessed or unwitnessed, non-traumatic, and non-drowning death in patients with epilepsy with or without evidence for a seizure, and excluding documented status epilepticus, in which postmortem examination does not reveal a structural or toxicological cause of death.[11]

Genetic evaluation was performed by Sanger sequencing according to physician criteria, phenotype suspicious and, in more recent cases, by a next-generation-sequencing (NGS) panel including 80 genes related to cardiac arrhythmia and SCD (S1 Table). Pathogenicity of identified genetic variants was established according to current recommendations[12], clinical evaluation and family pedigree. A positive NGS analysis was considered when a potential pathogenic (likely pathogenic or pathogenic) variant was present in the proband. On the other hand, a negative result from NGS analysis was considered when no potential pathogenic variant was detected. Finally, patients presenting a complex genotype (co-existence of two mutations in different genes) or a potential pathogenic variant in a rare gene (such as *CALM2*, *CASQ2* or *KCNJ2* genes) after NGS analysis, were considered in the "other group". Detailed information on genetic evaluation is included in S1 Methods.

## Statistical analysis

The data were analyzed with SPSS® software version 21.0 (Chicago, IL, USA). Results are expressed as mean plus or minus standard deviation (SD) or frequencies and percentages. Clinical characteristics were compared using $\chi^2$ or Fisher's exact test for categorical variables and unpaired Student-t test or Mann-Whitney U test for continuous variables. A *P* value of less than 0.05 was considered to be statistically significant. Variables related to SCA or SCD at first presentation were selected by univariable analysis (p <0.1) and contrasted by multivariable logistic regression with selection of variables through exclusion by steps.

## Results

Of the 50 index cases, 29 were male, with an age of 34.5±17.2 years. Median age at diagnosis was 32.5 years (12.7–43.2). Family history of SCD was present in 15 cases. Overall, 9 individuals experienced SCA/SCD and 18 cardiac syncope at the time of diagnosis. We identified a typical trigger in 31 probands, but the most prevalent circumstances of arrhythmic events were during rest and during exercise. The primary arrhythmia syndrome spectrum included LQTS (n = 19), BS (n = 15), CPVT (n = 15) and SQTS (n = 1). For 40 of 50 cases, genetic analyses were performed. For 10 BrS patients genetic evaluation was not performed based on absent of a positive SCD family history and the low yield of genetic testing in this syndrome. Overall,

genetic testing led by phenotype and physician criteria identified a disease-causing mutation in 31 of 40 patients. Distribution of genes was 8 of 50 for *RYR2* 6 of 50 for *KCNH2*, 5 of 50 for *KCNQ1*, 4 of 50 for *SCN5A* and 8 of 50 for other uncommon genotypes. Fig 1 represents phenotype-genotype distribution in all probands. Genotype details are added in the S2 Table.

Clinical features of the 50 cases, according to the previous diagnosis of epilepsy, are summarized in Table 1. Eight patients were evaluated for convulsive syncope and misdiagnosed with epilepsy before a channelopathy diagnosis was reached. All these patients with epilepsy misdiagnosis presented as generalized tonic-clonic seizures described in the in the emergency admission report as shock-like and irregular movements of both arms and legs. In a more detailed examination in the Inherited Cardiovascular Disease Clinic, it could be concluded that they had actually presented a cardiogenic syncope prior to seizures.

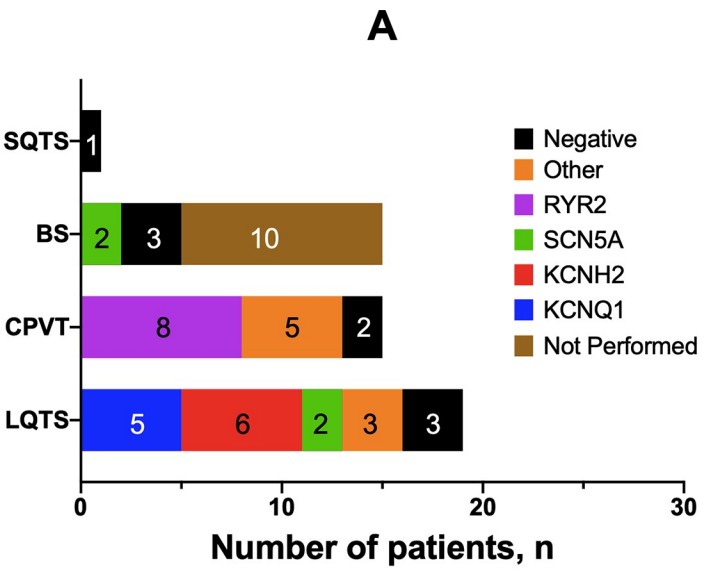

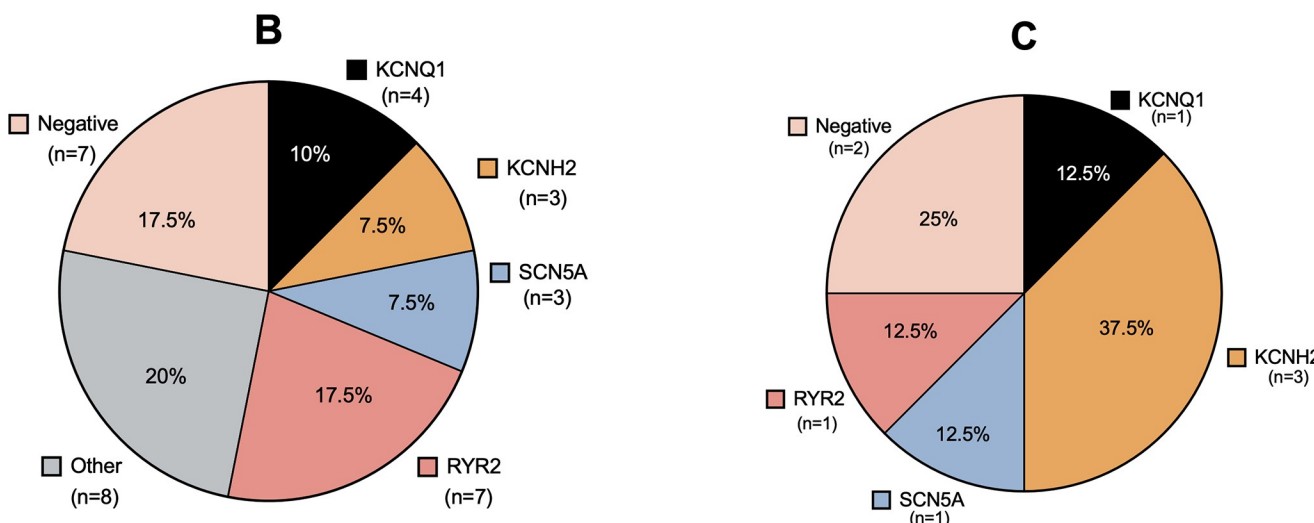

**Fig 1. Primary inherited arrhythmic syndromes and genotype distribution among all probands (panel A), probands without previous epileptiform diagnosis (panel B) and probands previously misdiagnosed (panel C).**

**Table 1. Clinical features of probands.** BS: Brugada syndrome; CPVT: catecholaminergic polymorphic ventricular tachycardia; ECG: electrocardiogram; FHx: family history; LQTS: long QT syndrome; SCA: sudden cardiac arrest; SCD: sudden cardiac death; SQTS: short QT syndrome.

| | Patients without previous epilepsy diagnosis (n = 42) | Patients with previous epilepsy diagnosis (n = 8) | *P* value |
|---|---|---|---|
| Male sex, n | 24 | 5 | .54 |
| Age at diagnosis (y), median (interquartile range) | 29.5 (12–41.5) | 41.5 (30.2–53) | .02 |
| Delay from first symptom (cardiac syncope or SCD) to (y) diagnosis, median (interquartile range) | 0.1 (1–1.15) | 18.5 (10.6–36.6) | < .0001 |
| SCD FHx, n | 12 | 3 | .61 |
| **Channelopathy diagnosis, n** | | | |
| BS | 14 | 1 | .23 |
| LQTS | 14 | 5 | .23 |
| SQTS | 1 | 0 | .84 |
| CPVT | 13 | 2 | .70 |
| **Genetic background, n** | | | |
| *SCN5A* | 3 | 1 | .51 |
| *KCNQ1* | 4 | 1 | .60 |
| *KCHN2* | 3 | 3 | .07 |
| *RYR2* | 7 | 1 | .62 |
| Others | 8 | 0 | .31 |
| Negative | 7 | 2 | .31 |
| Not performed | 10 | 0 | |
| **Triggers, n** | | | |
| Exercise | 9 | 3 | .37 |
| Emotion | 4 | 1 | .60 |
| Rest | 28 | 2 | .23 |
| Auditory stimuli | 1 | 2 | .29 |
| **Symptoms at diagnosis, n** | | | |
| Palpitations | 6 | 0 | .57 |
| Vasovagal syncope | 7 | 0 | .58 |
| Cardiogenic syncope | 14 | 4 | .43 |
| Other | 10 | 0 | .18 |
| SCA / SCD | 5 | 4 | .02 |
| **ECG, n** | | | |
| First ECG Diagnostic | 20 | 5 | .70 |

Regarding the EEG findings, there were no abnormalities in seven out of the eight patients; just in one patient (patient VI in Table 2) the EEG analysis showed "unspecific findings that could be related to Temporal Lobe Epilepsy", but a definitive diagnosis was not achieved. Further evaluations with neuroimaging techniques, as brain magnetic resonance, showed no abnormalities in the misdiagnosed patients. Clinical characteristics of these 8 patients are summarized in Table 2. All patients underwent neuroimaging and EEG, Median age of misdiagnosed patients with a seizure-related disorder was 41.5 years (30.2–53) and 5 were male. A final diagnosis of a channelopathy was made after recurrent cardiac syncope in 4 probands after SCA with electrocardiographically proven VF in 3, and post-mortem in 1. Three patients had a positive family history of SCD and as many as 6 presented a characteristic trigger of the sentinel events such as exercise, emotional stress, and auditory stimuli. There was a similar gender distribution between groups, but probands with epilepsy were significantly older. Presence of a positive family history of SCD was similar (Table 1 and Fig 2A).

**Table 2. Clinical features of patients with cardiac channelopathy and prior neurological diagnosis.** AED: anti-epileptic drug; BS: Brugada syndrome; CPVT: catechol-aminergic polymorphic ventricular tachycardia; ECG: electrocardiogram; LQTS: long QT syndrome; SADS: sudden arrhythmic death syndrome; SCA: sudden cardiac arrest; SCD: sudden cardiac death; SQTS: short QT syndrome; F: female; M: male; y: year.

| Patient | Sex | Age at diagnosis | Previous neurological diagnosis | Number of AEDs | Cardiac channelopathy | Gene test | Family history of SCD | Retrospective diagnostic first ECG | Trigger | Diagnostic delay (y) | Event at diagnosis |
|---------|-----|-----|-----|-----|-----|-----|-----|-----|-----|-----|-----|
| I | F | 22 | Epilepsy | 1 | CPVT | Negative | No | No | Exercise | 9.5 | Recurrent syncopes |
| II | M | 30 | Cryptogenic generalized epilepsy | 3 | LQTS2 | KCNH2 + | No | Yes | Auditory stimuli | 16.1 | SCA |
| III | M | 50 | Generalized epilepsy | 1 | BS | SCN5A + | No | No | Rest | 21 | Recurrent syncopes |
| IV | M | 39 | Epilepsy | 1 | LQTS | Negative | Yes | Yes | Emotional stress | 34.3 | Recurrent syncopes |
| V | M | 57 | Cryptogenic generalized epilepsy | 2 | LQTS1 | KCNQ1 + | Yes | Yes | Rest | 39 | Recurrent syncopes |
| VI | F | 44 | Temporal lobe epilepsy | 2 | LQTS2 | KCNH2 + | No | Yes | Auditory stimuli | 48.4 | SCA |
| VII | F | 31 | Generalized epilepsy | 3 | CPVT | RYR2 + | Yes | No | Exercise | 6.2 | SCA |
| VIII | M | 54 | Cryptogenic generalized epilepsy | 2 | LQTS2 | KCNH2 + | No | Yes | Exercise | 11.8 | SADS |

Patients with a prior epilepsy diagnosis constituted 5 of 19 for LQTS probands (3 *KCNH2* and 1 *KCNQ1*), 2 of 15 for CPVT probands (1 *RYR2*), and 1 of 15 for BS probands (1 *SCN5A*) (Fig 2B). Yield of genetic testing was high for both groups: 25 of 31 in probands without prior epilepsy diagnosis and 6 of 8 in patients with epilepsy diagnosis. Similarly, we did not find differences in distribution of disease-causing genes (Table 1 and Fig 1B), but there was a tendency towards a higher incidence of *KCHN2* variants in patients with a prior epilepsy diagnosis (OR 4.7, $P = .06$). At time of first ECG, the two groups presented similar diagnostic findings of particular channelopathies in each case ($P = .44$). The groups showed a comparable prevalence of cardiac channelopathy spectrum, but out of 8 misdiagnosed patients, 5 had LQTS. These patients all showed a QTc interval greater than 470 ms in the first ECG with a characteristic notched T wave (Fig 3); 4 of them were *KCNH2* mutation carriers.

Overall, the two groups presented a comparable distribution of identifiable event triggers (Fig 2C), but patients with a previous diagnosis of epilepsy were more likely to experience their sentinel arrhythmic event as an SCA/SCD episode (50% vs 10%; $P = .02$) (Fig 2D). For those misdiagnosed with epilepsy, the median age of channelopathy diagnosis was 41.5 years, 12 years later than those with a correct diagnosis from the beginning, and in those patients who presented cardiac syncope or SCD as first symptom, the median time from first symptom to diagnosis was extremely delayed, much more than in those without a epilepsy diagnosis (Table 1).

## Influence of misdiagnosis on clinical outcome

In patients with misdiagnosed epilepsy, being treated with two or more drugs (OR 8, $P = .029$) was significantly associated with SCA or SCD at presentation. Comparing both groups, univariable analysis revealed that a previous diagnosis of epilepsy (OR 6.6, $P = .02$), QTc of more than 460 ms at first event (OR 6, $P = .02$), and presence of *KCNH2* mutation (OR 4.7, $P = .06$)

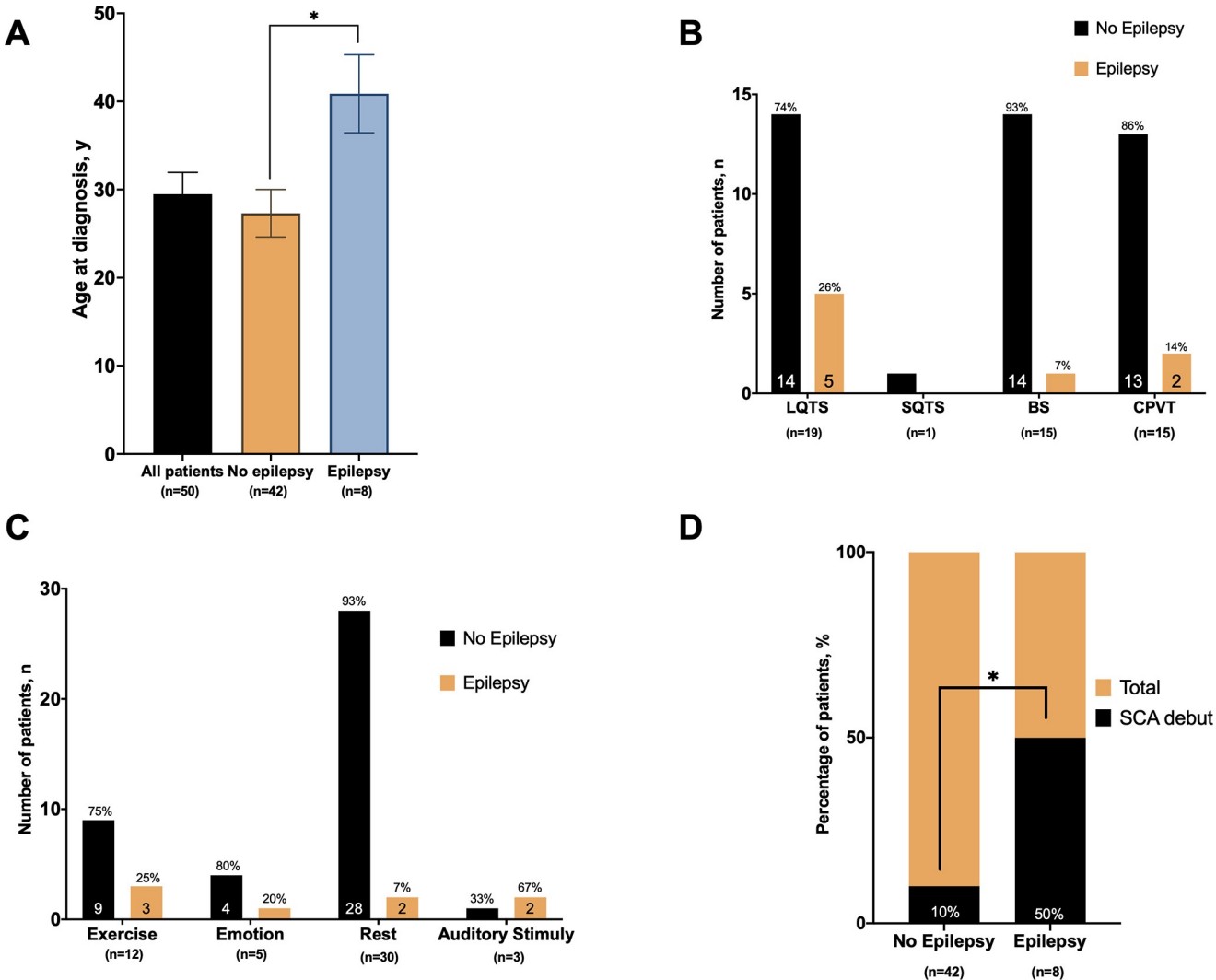

**Fig 2.** Age at diagnosis in years in the two different groups showing a significantly delayed diagnosis in the previously misdiagnosed patients (Panel A). Compared distribution of primary inherited arrhythmic syndromes among probands (Panel B). Distribution of typical triggers within probands (Panel C). Percentage of patients presented with sudden cardiac arrest at time of diagnosis (Panel D). *number above bars represent the percentage among the group. †* means significance p value <0.05.

were significantly associated with (or showed a tendency towards) SCA or SCD at first presentation. After multiple logistic regression analysis we identified epilepsy misdiagnosis as an independent predictor of an SCA or SCD event at time of channelopathy diagnosis (OR 7.46, *P* = .03 (Table 3).

## Discussion

Previously reported series demonstrated that individuals with epilepsy have an increased risk of sudden unexpected death, especially in the young.[13] This tragic event is often referred to as sudden unexpected death in epilepsy, or SUDEP.[11,14] Cardiac arrhythmias due to unrecognized cardiac channelopathies have been described as one of the possible underlying causes that might provoke SUDEP.[15] The diagnostic error usually occurs in patients presenting with arrhythmic syncope and a secondary convulsive episode caused by brain hypoxia, but the

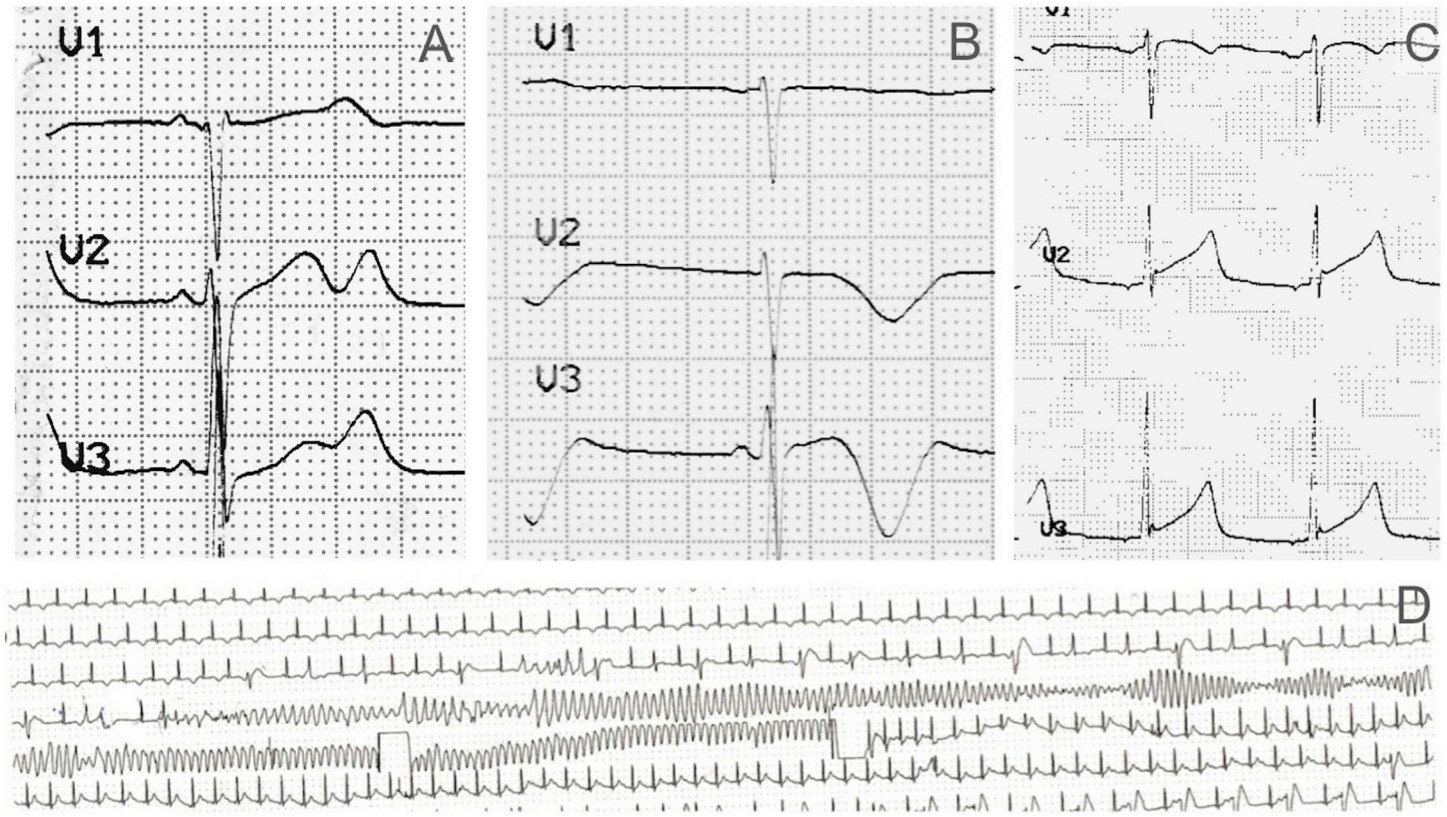

**Fig 3. Representative ECG traces of misdiagnosed probands, showing typical LQTS features.** Panels A to C show basal ECG from three *KCNH2* mutation carriers with extremely prolonged QT intervals and abnormal T-wave shape (A: notched T wave, B: inverted and wide T wave, and C: widened T wave). Panel D shows an ECG recording during one of the syncopal attacks with convulsive status suffered by the patient in panel A, who corresponds to patient II in Table 2, demonstrating a torsade de pointes episode that required external defibrillation.

central issue is low cardiac output due to a polymorphic ventricular arrhythmia.[16] Whether clinical misdiagnosis of epilepsy has significant implications for patient outcomes has not been defined before. In this single-center cohort of 50 patients with a clinical diagnosis of probable or definite cardiac ion channelopathy, we show that this epilepsy misdiagnosis has a negative impact on clinical outcomes of patients with underlying channelopathies, emphasizing the need for accurate cardiological evaluation in cases with syncope or seizures of unclear origin.

Real etiology for this sudden death cases still remains a matter of debate. Risk factors for this tragic event among patients with a clear diagnosis of epilepsy include young age and frequent generalized tonic-clonic seizures[17]. However, up to 20% epilepsy diagnosis, according to previous data, are wrong due to an underlying cardiovascular condition[18]. Whether all SUDEP cases are caused by this misdiagnosis of an underlying cardiac channelopathy, as a result of malignant ventricular arrhythmias, cannot be concluded from our work. Recent

**Table 3. Univariable and multivariable predictors.** QTc: corrected QT interval.

| Predictors | Univariate analysis OR | *P* value | Multivariate analysis OR | *P* value |
|---|---|---|---|---|
| **Prior epilepsy diagnosis** | 6.6 | .02 | 7.46 | .03 |
| **QTc > 460 ms** | 6 | .02 | | NS |
| ***KCNH2* mutation** | 4.7 | .06 | | NS |

studies have deeply investigated the pathophysiology of SUDEP, and there is agreement that cardiac dysfunction plays a major role.[19] These studies have demonstrated that a severe depression of the cardiac and respiratory centers may arise after a generalized tonic-clonic seizure, particularly among young patients, and this fact can provoke severe hypoxemia and/ or hypotension, leading to an instant death[20]. On the other hand, among cardiac dysfunction causes there is lack of evidence for the existence of malignant ventricular arrhythmias in the context of epilepsy.[21–22] Similarly to our findings, McCormick et al[6] reported that some patients were classified as epileptic for many years until a final LQTS diagnosis was reached. Some of that patients suffered serious arrhythmias during follow up, highlighting the needing of a high degree of suspicious when facing patients with a not completely typical epilepsy.

Another possible link between these two apparently unrelated entities is the presence of common ion channels that are present either in central nervous system or in cardiac conduction system. In fact, some mutations associated with epilepsy have been identified in cardiac arrhythmia genes, like SCN5A, KCNH2 and KCNQ1[23–25]. These data have not been validated in larger cohorts, and the hypothesis remains controversial. There is evidence of patients with coexistence of epilepsy and cardiac arrhythmias with a single mutation, but definitive clinical or functional data demonstrating a unique cardio-cerebral channelopathy are absent. [26] In our study, *KCNH2* mutations showed no statistically significant relationship with the previous diagnosis of epilepsy. *KCNH2* mutation carriers are at increased arrhythmic risk compared to other gene loci, particularly for mutations at the pore[27]; in our families genotype-phenotype segregation was observed, with no evidence for neurological impairment in any of the carriers. In fact, in our cohort, after thorough review, none of the misdiagnosed cases showed typical traces of epilepsy, as neurological imaging exams and EEGs were normal. Furthermore, a clear diagnosis of channelopathy was patent in ECG traces so our data do not support the shared cardio-cerebral channels hypothesis.

Sudden unexpected death in epilepsy, which is likely underestimated, usually occurs during the night.[28] These data are different from our series, where episode triggers were more similar to those observed for arrhythmia in cardiac channelopathies, such as exercise, auditory stimuli, and adrenergic situations. The presence of this clinical context should be a warning sign for the presence of a cardiac channelopathy such as LQTS or CPVT. Moreover, a family history of sudden death, which was present in more than a third of the misdiagnosed patients in our cohort, should raise suspicions of the underlying genetic cardiac condition. Finally, correct interpretation of ECG traces in doubtful epilepsy cases is critical, especially in patients with normal neurological tests and poor response to anticonvulsant treatment; 6 out of 8 patients showed clear evidence of LQTS or BS in the ECG. Besides, comparison of ECG features between cases with correct and incorrect diagnosis was similar, suggesting an interpretation mistake. This confirms the need for close collaboration between neurologists and cardiologists to avoid diagnostic delay when facing with atypical epileptic cases, syncope, or SUDEP.[29]

Clinical outcome was poor in misdiagnosed cases. McCormick et al study reported that LQTS patients labeled as epileptic experienced a particularly long diagnostic delay and that ECGs were frequently requested but interpretation errors were common.[6] Our data develop and extend this demonstration of clinical impact: there was not only a long diagnostic delay in misdiagnosed cases, but also a higher incidence of VF/SCD. Syncope and absence of medical therapy (particularly betablockers in LQTS and CPVT cases) are strong predictors of impaired survival in cardiac channelopathies.[10] These two features were present in all the misdiagnosed cases. Of particular interest is the finding that the use of more than 1 AED is an independent predictor of adverse outcome, given that some of these drugs have a potential arrhythmogenic

role in LQTS due to their ability to prolong the action potential and the QT interval. These results have a strong clinical impact in daily practice.

As a retrospective study, it is susceptible to many limitations. First, although neurological medical records were rigorously evaluated, the analyzed data may be incomplete in some patients. Second, genetic evaluation was not performed in all probands and this could underestimate genetic testing yield. Third, patients are a highly selected population from a single referral center limiting the representation of these patients. Finally, the sample size is modest, albeit for an uncommon clinical entity such as cardiac channelopathies results can be clinically interpreted with enough accuracy.

## Conclusion

Cardiac channelopathy patients can be misdiagnosed with epilepsy. This involves diagnostic delay, prolonged absence of correct antiarrhythmic therapy, and the use of AEDs with a potential proarrhythmic effect, resulting in increased risk of malignant ventricular arrhythmias and SCD.

## Supporting information

**S1 Methods. Detailed genetic evaluation.**
(DOCX)

**S1 Table. Genes included in the NGS panel.**
(DOCX)

**S2 Table. Mutations identified in patients diagnosed with a cardiac channelopathy.**
(DOCX)

**S1 File. Database with anonymized patients data, for independent replication of the results.**
(SAV)

## Author Contributions

**Conceptualization:** Miguel Álvarez López, Luis Tercedor Sánchez, Juan Jiménez-Jáimez.

**Data curation:** Javier Ramos-Maqueda, Rosa Macías Ruiz, Mercedes Cabrera Ramos, Manuel Molina Lerma, Pablo Sánchez Millán, Juan Jiménez-Jáimez.

**Formal analysis:** Francisco Bermúdez-Jiménez, Manuel Molina Lerma, Pablo Sánchez Millán.

**Investigation:** Luis Tercedor Sánchez, Juan Jiménez-Jáimez.

**Methodology:** Javier Ramos-Maqueda, Mercedes Cabrera Ramos, Miguel Álvarez López, Juan Jiménez-Jáimez.

**Supervision:** Francisco Bermúdez-Jiménez, Rosa Macías Ruiz, Juan Jiménez-Jáimez.

**Validation:** Rosa Macías Ruiz, Manuel Molina Lerma.

**Writing – original draft:** Javier Ramos-Maqueda, Francisco Bermúdez-Jiménez, Juan Jiménez-Jáimez.

**Writing – review & editing:** Luis Tercedor Sánchez, Juan Jiménez-Jáimez.

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
