## [Decision Letter · Decision Letter 0]

31 Dec 2019

PONE-D-19-31902

Prognostic impact of misdiagnosis of cardiac channelopathies as epilepsy

PLOS ONE

Dear Dr Jiménez-Jáimez

Thank you for submitting your manuscript to PLOS ONE. After careful consideration, we feel that it has merit but does not fully meet PLOS ONE’s publication criteria as it currently stands. Therefore, we invite you to submit a revised version of the manuscript that addresses the points raised during the review process.

The editor and reviewers have carefully gone through your manuscript and found it interesting, but there are several concerns that need to be revised.  Please respond each criticism raised up the reviewers clearly.

We would appreciate receiving your revised manuscript by Feb 07 2020 11:59PM. To enhance the reproducibility of your results, we recommend that if applicable you deposit your laboratory protocols in protocols.io, where a protocol can be assigned its own identifier (DOI) such that it can be cited independently in the future. For instructions see: http://journals.plos.org/plosone/s/submission-guidelines#loc-laboratory-protocols

We look forward to receiving your revised manuscript.

Kind regards,

Katriina Aalto-Setala, Professor

Academic Editor

PLOS ONE

Journal Requirements:

3. Please discuss the limitations of your study in the discussion section of your manuscript.

4. Thank you for submitting the above manuscript to PLOS ONE. During our internal evaluation of the manuscript, we found significant text overlap between your submission and the following previously published works, mainly in the discussion section.

https://www.mdpi.com/1422-0067/20/8/1979

Please revise the manuscript to rephrase the duplicated text, cite your sources, and provide details as to how the current manuscript advances on previous work. Please note that further consideration is dependent on the submission of a manuscript that addresses these concerns about the overlap in text with published work.

Reviewers' comments:

Reviewer's Responses to Questions

**Comments to the Author**

1. Is the manuscript technically sound, and do the data support the conclusions?

Reviewer #1: Yes

Reviewer #2: Yes

Reviewer #3: Partly

2. Has the statistical analysis been performed appropriately and rigorously? 

Reviewer #1: Yes

Reviewer #2: Yes

Reviewer #3: Yes

3. Have the authors made all data underlying the findings in their manuscript fully available?

Reviewer #1: Yes

Reviewer #2: Yes

Reviewer #3: Yes

4. Is the manuscript presented in an intelligible fashion and written in standard English?

Reviewer #1: Yes

Reviewer #2: Yes

Reviewer #3: Yes

5. Review Comments to the Author

Reviewer #1: The authors report their experience about fifty families with channellopathy and they collected information about their epilepsy. This study focuses attention on a very interesting topic. The manuscript has some inaccuracies that I have detailed below:

- Population and study Group: the weak point of this part is the selection of the patients: I suggest it could be more helpful if the authors mentione more about the inclusion and exclusion criteria for the enrollment of the patients.

- Genetic evaluation: this section must be expanded because more details about the methods used must be reported.

- Results: it is essential to describe the clinical symptoms and EEG abnormalities found in the patients in order to support the main message of the study. Please add these details.

- Discussion: The discussion of the manuscript is inconclusive; the authors must discuss better the genetic of sudden unexpected death (see the paper by Manolis TA, et al. Seizure. 2019;64:65-73). Moreover, the authors must dedicate more room to the possible link between cardiac channellopathies and epilepsy (see the papers by Li MCH, et al. Epilepsia. 2019;60:1753-1767 and by Iannetti P et al. Eur Rev Med Pharmacol Sci. 2017;21:5523-55269). All these references should be quoted and briefly but incisively discussed.

In summary, the paper would acquire greater interest and attraction by giving emphasis to the possible genetic links between cardiac and neurological channellopathies.

Reviewer #2: This is a well written and clear manuscript with a clinically important finding about the significance of misdiagnosis in Long QT syndrome. Although the study group is not large the central finding has not been examined in other larger published Long QT cohorts, and there is only one previous comparable paper in the area from 2009.

The paper would benefit from some revisions, Major:

Discussion - although the paper is concerned principally with the the misdiagnosis of inherited cardiac channelopathies as epilepsy a substantial early part of the discussion (para 2) focuses on SUDEP - the occurrence of sudden death in individuals with a correct diagnosis of epilepsy. This includes a consideration of whether cardiac channels are also expressed in the CNS and contribute to epilepsy. This seems to be perpetuating the confusion that the paper is aiming to clarify - namely that these patients didn't have epilepsy and that this misdiagnosis was evident in the majority of cases from their first ECG.

I recommend that the discussion of SUDEP should be given less emphasis and the more relevant discussion relating to the clinical distinction between cardiac syncope and true seizures be expanded.

Logistic regression - the information on the regression analysis is not well presented and more detail is needed. What was the full list of variables considered? Age at diagnosis appears to be an important difference between the two groups, were the influence age and gender examined? Treatment with >1 AED was the most significant variable in the univariate analysis but it is not included in table 3 and no mention is made of what happened to this variable in the multiple logistic regression.

Minor:

Intro para 2: Therefore, patients at risk of SCD... suggest: As a result patients at risk of SCD...

Results para 2: "All patients underwent neuroimaging and EEG, showing no definite evidence of neurological disease" - this statement is vague. Were all the investigations actually normal? If not please describe any abnormalities.

Table 1 "First diagnostic ECG" - better labelled, 'First ECG diagnostic'

Discussion - last para - discussion of AE Drugs. Do you have any information about the specific drugs that individuals were on at the time of the SCA/SCD events? Were they on known QT prolonging meds?

Supplementary table S1 - please provide transript details (i.e. NM_ and NP_ numbers) that you have used to annotate these variants and ensure that you are using the standard (cardiac) transcripts. For instance Patient 2 is annotated form NM_000238.3, which is standard for KCNH2 but patient 1 is annotated from a different transcript.

It isn't clear to me how some of these variants reach a classification of likely pathogenic/pathogenic on standard ACMG criteria. For instance, based on current information our centre would only classify KCNH2 I30F as a VUS. Can you provide more detail about your classification pathway?

Reviewer #3: The topic of the paper (possible misdiagnosis of epilepsy in the context of cardiac channelopathy) is timely and of interest. The problem with the manuscript is the very modest number of subjects studied and some inaccuracies in the presentation of the numbers.

General comments:

The material consists of 50 patients (as such, quite limited number), and genetic analyses were performed only in 40 cases. Moreover, DNA diagnosis could be established in only 31 of these 40 cases, leaving 10 + 9 cases somewhat unkown in nature. The authors should seriously take these figures into account when they sum their conclusions e.g. on the increased risk of SCA/SD. One may even ask whether they should collect more cases before publication. The least they need to do is to provide a critically written paragraph "Study limitations" at the end of Discussion.

Specific comments:

1. It is irritating to give percentage values with decimals when analysing numbers of <40, in particular those of <8 (epilepsy cases). Please correct.

2. Fig. 1: Does panel A, combined with the text, indeed show that 10 patients with Brugada syndrome could not be verified with DNA? This should be explained if this is the case.

3. There appears to be no legend for Fig. 2D.

4. Results, 1st para: "Distribution of genes was..." suggests 8 + 6 + 5 + 4 = 23 but the previous sentence tells that there 31 disease-causing mutations. Reason for this discrepancy?

5. Table 1: Again, there is some problem with the n values. At "Genetic background" the numbers and %values are OK for the epilepsy cases (n = 8) but remain obscure for the other group (n = 31). It is impossible to get the sum 31 by any inspection of the data. And what is the difference between "Negative" and "Others" in this portion of the Table? My concern is that if there is some confusion with the figures, should there perhaps also be concern on the conclusions drawn?

6. PLOS authors have the option to publish the peer review history of their article (what does this mean?). If published, this will include your full peer review and any attached files.

Reviewer #1: Yes: Alberto Verrotti

Reviewer #2: No

Reviewer #3: No

---

## [Author Response · Author response to Decision Letter 0]

17 Feb 2020

Editors Commets

Journal Requirements:

Response: Thanks for your comment. We have adressed these additional points to assure a complete adherence to Plos One style.

Response: we have included captions for Sup. Files at the end of the manuscript, and updated the text citations.

b) If there are no restrictions, please upload the minimal anonymized data set necessary to replicate your study findings as either Supporting Information files or to a stable, public repository and provide us with the relevant URLs, DOIs, or accession numbers. Please seehttp://www.bmj.com/content/340/bmj.c181.long for guidelines on how to de-identify and prepare clinical data for publication. For a list of acceptable repositories, please see http://journals.plos.org/plosone/s/data-availability#loc-recommended-repositories.

Response: Dear Editor. There are no restrictions to access the minimun anonimyzed data set. We have uploaded, as Supplementary file, the database with anonimyzed patients data, for independent replication of our results

4. Please discuss the limitations of your study in the discussion section of your manuscript.

Response: we have added a limitation paragraph.

5. Thank you for submitting the above manuscript to PLOS ONE. During our internal evaluation of the manuscript, we found significant text overlap between your submission and the following previously published works, mainly in the discussion section.

https://www.mdpi.com/1422-0067/20/8/1979

Please revise the manuscript to rephrase the duplicated text, cite your sources, and provide details as to how the current manuscript advances on previous work. Please note that further consideration is dependent on the submission of a manuscript that addresses these concerns about the overlap in text with published work.

Response: Dear Editor. It was not our aim to literally replicate any text from another publication. Certainly, we have read with interest the paper you mention by Coll et al, and some of our data are concordant. Perhaps we have involuntarily reproduced some similar information from this paper., We have carefully reviewed the discussion and reformulated some very similar phrases. We are very thankful for your thorough review, as this is something we had completely missed.

 

Reviewer #1: The authors report their experience about fifty families with channellopathy and they collected information about their epilepsy. This study focuses attention on a very interesting topic. The manuscript has some inaccuracies that I have detailed below:

- Population and study Group: the weak point of this part is the selection of the patients: I suggest it could be more helpful if the authors mentione more about the inclusion and exclusion criteria for the enrollment of the patients.

Response: Dear reviewer, thank you for your comment. Clearly it helps to improve the manuscript. Information regarding criteria for considering the presence or absence of epilepsy have been added in the method section (population and study groups). For considering an epilepsy diagnosis we contemplated the presence of either a personal history of seizures, epilepsy or a history of AED therapy after a neurologist assessment, including EEG and neuroimaging. Furthermore, we provide exclusion criteria among probands. We excluded all acquired causes of seizures comprising traumatic/vascular injury, fever or metabolism disbalance.

- Genetic evaluation: this section must be expanded because more details about the methods used must be reported.

Response: Thanks for your comment. We are aware that genetic evaluation methodology was succinctly described, mainly due to words account restriction. Methods on genetic evaluation have been briefly expanded in the main manuscript and further description is available in the supplementary data. We also attach a table including all genes analyzed in the NGS panel (Table 1 in supplementary material). The vast majority of genetic tests were performed by a multidisciplinary and experienced team heading by Dr. Lorenzo Monserrat, and finally interpreted in the patient/family context by our group.

- Results: it is essential to describe the clinical symptoms and EEG abnormalities found in the patients in order to support the main message of the study. Please add these details.

Response: According to the emergency admission reports, all the patients misdiagnosed with epilepsy presented generalized tonic-clonic seizures, described as shock-like and irregular movements of both arms and legs. 

After a thorough evaluation in the Inherited Cardiovascular Disease Clinic and a primary arrhythmia syndrome was diagnosed, we can suspect that patients presented a cardiogenic syncope prior to seizures, and that these could be a consequence of the cerebral hypoxia that occurs in syncopes.

- Discussion: The discussion of the manuscript is inconclusive; the authors must discuss better the genetic of sudden unexpected death (see the paper by Manolis TA, et al. Seizure. 2019;64:65-73). Moreover, the authors must dedicate more room to the possible link between cardiac channellopathies and epilepsy (see the papers by Li MCH, et al. Epilepsia. 2019;60:1753-1767 and by Iannetti P et al. Eur Rev Med Pharmacol Sci. 2017;21:5523-55269). All these references should be quoted and briefly but incisively discussed.

In summary, the paper would acquire greater interest and attraction by giving emphasis to the possible genetic links between cardiac and neurological channellopathies.

Response: Thanks for this useful comment. As the reviewer states, etiology of SUDEP remains a matter of debate. It´s likely that there is not a single explanation for these unexplained deaths. We agree genetics play a role in SUDEP cases, but there is some heterogeneity and many doubts about the real cause in some of SUDEP cases. Our work focuses in those cases with a clear underlying cardiac genetic condition. Whether always there is a pitfall in diagnosis or there is a true cardio-cerebral channelopathy cannot be concluded from our data. In the revised version of the manuscript, we have gone deeper inside this issue, reviewing and quoting the interesting references the reviewer suggested. The article by Manolis TA et al is a thorough review about this possible cardio-neural connection, suggesting but not confirming the co-existence of both disorders. They suggest that many of SUDEP cases are really caused by this neuro-cardio-respiratory disfunction that leads to severe cardiorespiratory depression after a tonic-clonic generalized seizure. Our work focuses in a different clinical scenario, with a not definitive epilepsy diagnosis in all cases, so we cannot fully support this cardio-cerebral syndrome with our data. This is in agreement with the interesting editorial you have suggested, by Ianneti. There is not definitive evidence for the real co-existence of brain and Heart channelopathies, in the same patients, and for the same mutations. We have re-formulated this issues and cited these suggested references.

 

Reviewer #2: This is a well written and clear manuscript with a clinically important finding about the significance of misdiagnosis in Long QT syndrome. Although the study group is not large the central finding has not been examined in other larger published Long QT cohorts, and there is only one previous comparable paper in the area from 2009.

The paper would benefit from some revisions, Major:

Discussion - although the paper is concerned principally with the the misdiagnosis of inherited cardiac channelopathies as epilepsy a substantial early part of the discussion (para 2) focuses on SUDEP - the occurrence of sudden death in individuals with a correct diagnosis of epilepsy. This includes a consideration of whether cardiac channels are also expressed in the CNS and contribute to epilepsy. This seems to be perpetuating the confusion that the paper is aiming to clarify - namely that these patients didn't have epilepsy and that this misdiagnosis was evident in the majority of cases from their first ECG.

I recommend that the discussion of SUDEP should be given less emphasis and the more relevant discussion relating to the clinical distinction between cardiac syncope and true seizures be expanded.

Response: we are thankful for this useful comment, and we certainly agree in your statement. We are aiming to describe the poor clinical outcome in cases of underlying real channelopathy with no evidence of definitive epilepsy. We have reformulated the discussion giving more emphasis to cardiac conditions. We have not completely deleted other possible explanations for SUDEP as the cardiac autonomic system depression or the cardio-cerebral syndrome as we feel that they might explain some SUDEP cases, but we made a statement that our work do not support this hypothesis, albeit we cannot fully exclude it.

Logistic regression - the information on the regression analysis is not well presented and more detail is needed. What was the full list of variables considered? Age at diagnosis appears to be an important difference between the two groups, were the influence age and gender examined? Treatment with >1 AED was the most significant variable in the univariate analysis but it is not included in table 3 and no mention is made of what happened to this variable in the multiple logistic regression.

Response: Dear reviewer, the variables considered for the multivariate analysis were those which showed a tendency towards (p<0.1) or were significantly associated (p<0.005) with SCA or SCD at first presentation. Age at diagnosis was different in both groups because an epilepsy misdiagnosis led to a delay of the channelopathy diagnosis in this group. However, it was not associated with an increased risk of SCA or SCD event at time of channelopathy diagnosis (p=0.57) just as it wasn't the gender (0.78). 

So both were considered for statistical analysis and none of them was were associated with an increased risk of SCA or SCD event at time of channelopathy diagnosis.

Treatment with >1 AED is a variable just present in those patients with epilepsy misdiagnosis. Therefore, we performed an univariate analysis in the patients (n=8) who were treated with AED, that is a completely different analysis that the univariate analysis that we perform with the variables present in all patients (n=50) so it could not be included in the multivariate analysis performed with those variables which showed a tendency towards (p<0.1) or were significantly associated (p<0.005) with SCA or SCD at first presentation in the group of 50 patients.

Minor 

Intro para 2: Therefore, patients at risk of SCD... suggest: As a result patients at risk of SCD...

Response: Thank you for your suggestion, we changed the sentence.

Results para 2: "All patients underwent neuroimaging and EEG, showing no definite evidence of neurological disease" - this statement is vague. Were all the investigations actually normal? If not please describe any abnormalities.

Response: Dear reviewer, as it is described in result section, there were no relevant abnormalities at EEG evaluation. However, in one patient (patient VI in table 2) the EEG analysis showed “unspecific findings that could be related to Temporal Lobe Epilepsy”, bur not a definite diagnosis was performed . This information is reflected in table 2. Further evaluations with neuroimaging techniques, as brain magnetic resonance, showed no abnormalities in none of the misdiagnosed patients

Table 1 "First diagnostic ECG" - better labelled, 'First ECG diagnostic'

Response: The sentence has been changed, as suggested by the reviewer.

Discussion - last para - discussion of AE Drugs. Do you have any information about the specific drugs that individuals were on at the time of the SCA/SCD events? Were they on known QT prolonging meds?

Response: Dear reviewer, six of the eight patients were on valproic acid, two patients were on carbamazepine and lamotrigine, respectively. Until now, there is no evidence for an increased risk of acquired QT prolongation (https://crediblemeds.org).

This is an important issue of our paper. Here we point that not only misdiagnosed patients are at risk of taking potentially hazardous drugs, but not responding to one AEDs or the need for several AEDs may arise the suspicion of an underlying cardiac channelopathy. This fact was independently associated with an adverse outcome.

Supplementary table S1 - please provide transript details (i.e. NM_ and NP_ numbers) that you have used to annotate these variants and ensure that you are using the standard (cardiac) transcripts. For instance Patient 2 is annotated form NM_000238.3, which is standard for KCNH2 but patient 1 is annotated from a different transcript. It isn't clear to me how some of these variants reach a classification of likely pathogenic/pathogenic on standard ACMG criteria. For instance, based on current information our centre would only classify KCNH2 I30F as a VUS. Can you provide more detail about your classification pathway?

Response: Dear reviewer, genetic analysis were performed in a worldwide recognized center in cardiovascular genetics (Health in Code, Dr. Lorenzo Monserrat). 

These analyisis are performed by a multidisciplary team, according to the guidelines of EuroGentest and the American College of Medical Genetics (ACMG) appropriately with the requirements of the UNE-EN ISO 15189 and CLIA-88 standards as quality standards for clinical laboratories.

The potential pathogenicity of the identified variants in probands is initially evaluated based on previous description in clinical literature, as well as published in-vitro or in-vivo studies and bioinformatics. This information is analyzed and interpreted by experienced cardiologists and geneticists. Further information regarding genetic evaluation is available in the supplemental material.

Finally the information from genetic analysis is evaluated in the context of the patient’s clinical scenario and family screening. For example the I30F KCNH2 you mentioned, was interpreted as likely pathogenic based on previously described pathogenic variants affecting the same aminoacidic residue and gene region (PAS domain), as well as, bioinformatics predictors. Proven cosegregation in this extensive family supported its pathogenicity. We provide the family pedigree and ECG of three cases from two different generations.

Reviewer #3: The topic of the paper (possible misdiagnosis of epilepsy in the context of cardiac channelopathy) is timely and of interest. The problem with the manuscript is the very modest number of subjects studied and some inaccuracies in the presentation of the numbers.

Response: we are thankful for your positive comments. We agree the numbers are not high. We are dealing with uncommon clinical entities such as cardiac channelopathies and 50 seems to be an acceptable number to get some conclusions. However, we have added a limitation statement to accept this limitation, and all your considerations about the number presentations and statistics have been taken into account.

General comments:

The material consists of 50 patients (as such, quite limited number), and genetic analyses were performed only in 40 cases. Moreover, DNA diagnosis could be established in only 31 of these 40 cases, leaving 10 + 9 cases somewhat unkown in nature. The authors should seriously take these figures into account when they sum their conclusions e.g. on the increased risk of SCA/SD. One may even ask whether they should collect more cases before publication. The least they need to do is to provide a critically written paragraph "Study limitations" at the end of Discussion.

Response: we agree with this limitation. In this unicentric cohort of patients it´s unlikely to be able to add more patients. We feel statistics can help us understand and conclude some interesting points in studies with such limitation. In our opinion, and adding your suggested limitation paragraph, we feel these data are of clinical value and help the reader understand the poor clinical outcome in cases of clear cardiac channelopathy, but wrongly labelled as epileptic. 

Specific comments:

1. It is irritating to give percentage values with decimals when analysing numbers of <40, in particular those of <8 (epilepsy cases). Please correct.

Response: Thank you for your suggestion, we changed the numbers presentation in the manuscript and tables.

2. Fig. 1: Does panel A, combined with the text, indeed show that 10 patients with Brugada syndrome could not be verified with DNA? This should be explained if this is the case.

Response: Dear reviewer, as you correctly remark, genotype analysis of 10 Brugada Syndrome (BrS) patients were not performed. 

As previously reported in the HRS/EHRA Expert Consensus Statement on the State of Genetic Testing for the Channelopathies and Cardiomyopathies (Ackerman MJ, 2011), the diagnosis of BrS is clinical: based on ECG and clinical presentation, however genetic evaluation is not involved.

Since genetic evaluation is not implicated for neither diagnosis or risk stratification, in the context of the absence of a positive family history of SCD, we do not routinely perform genetic evaluation. It could lead to misinterpretations (Risgaard B; Clin Genet 2013) and unjustificable alarm and morbidity associated with unnecessary medical interventions among patients and relatives.

Moreover, although genetic testing costs are decreasing and is widely available, genetic testing yield in BrS is very low, about 30% (Priori SG. Circulation, 2002), limiting its benefice. Based on this reasons, we feel that genetic testing in BrS is recommended in selected patients. We have included and explained this point in results section, and added the information in Figure 1.

3. There appears to be no legend for Fig. 2D.

Response: Thank you for your comment, it was not properly indicated. It has been corrected.

4. Results, 1st para: "Distribution of genes was..." suggests 8 + 6 + 5 + 4 = 23 but the previous sentence tells that there 31 disease-causing mutations. Reason for this discrepancy?

Response: Thank you for your observation. In order to expose a clearer presentation of the data, we described the most common genes in this peculiar population. Certainly, probands with a positive genotype are 31:

8 RYR2 + 6 KCNH2 + 4 SCN5A + 5 KCNQ1 + 8 OTHERS = 31. “Others” group includes probands with mutation in CALM2 (protein calmodulin 2) (2 probands) (Jiménez-Jáimez J. PLoS One. 2016), CASQ2 (protein calsequestrin 2) (3 probands) and KCNJ2 (protein Kir2.1) (2 probands), responsible for the development of CPVT and Andersen-Tawil Syndrome, respectively. Furthermore one patient carrying pathogenic variants in KCH2 and KCNQ1 genes was considered in the group “others”.

This “others” group has been added to the revised main manuscript (results, 1st para). Furthermore, genotype details from the 31 probands are available in the supplementary material.

5. Table 1: Again, there is some problem with the n values. At "Genetic background" the numbers and %values are OK for the epilepsy cases (n = 8) but remain obscure for the other group (n = 31). It is impossible to get the sum 31 by any inspection of the data. And what is the difference between "Negative" and "Others" in this portion of the Table? My concern is that if there is some confusion with the figures, should there perhaps also be concern on the conclusions drawn?

Response: Dear reviewer, since “others” group was not properly described in the manuscript, information in text and tables was confusing. Thank you for your comment, we have improved the presentation of our data. 

In table 1, there were various errors in the numbers, I would like to deeply apologize for that. These mistakes have been resolved and we have also added “not performed” row for a clearer presentation of data. 

For clarification, a negative result from NGS analysis was considered when no potential pathogenic variant was detected. On the other hand, patients who presented a complex genotype (multiple mutations in different genes) or a potential pathogenic variant in a rare gen (in our case, CASQ2, CALM2 and KCNJ2) after NGS analysis, were considered in the “other group”. 

Furthermore, in the “method/5 para” and “results/1st para” sections, information regarding genotype analysis has been extended including, and brief description of criteria for patients distribution among groups of analysis.

Thankfully, conclusions are based on the numbers presented in the manuscript text and this confusion with presentation of numbers in the table does not change the conclusions and key messages from our study.

---

## [Decision Letter · Decision Letter 1]

25 Mar 2020

Prognostic impact of misdiagnosis of cardiac channelopathies as epilepsy

PONE-D-19-31902R1

Dear Dr. Jimenez-Jaimez,

We are pleased to inform you that your manuscript has been judged scientifically suitable for publication and will be formally accepted for publication once it complies with all outstanding technical requirements.

With kind regards,

Katriina Aalto-Setala, Professor

Academic Editor

PLOS ONE

Additional Editor Comments (optional):

Reviewers' comments:

Reviewer's Responses to Questions

**Comments to the Author**

1. If the authors have adequately addressed your comments raised in a previous round of review and you feel that this manuscript is now acceptable for publication, you may indicate that here to bypass the “Comments to the Author” section, enter your conflict of interest statement in the “Confidential to Editor” section, and submit your "Accept" recommendation.

Reviewer #1: All comments have been addressed

Reviewer #2: All comments have been addressed

Reviewer #3: All comments have been addressed

2. Is the manuscript technically sound, and do the data support the conclusions?

Reviewer #1: Yes

Reviewer #2: (No Response)

Reviewer #3: Yes

3. Has the statistical analysis been performed appropriately and rigorously? 

Reviewer #1: Yes

Reviewer #2: (No Response)

Reviewer #3: Yes

4. Have the authors made all data underlying the findings in their manuscript fully available?

Reviewer #1: Yes

Reviewer #2: (No Response)

Reviewer #3: Yes

5. Is the manuscript presented in an intelligible fashion and written in standard English?

Reviewer #1: Yes

Reviewer #2: (No Response)

Reviewer #3: Yes

6. Review Comments to the Author

Reviewer #1: After all changes made, in my opinion, the quality of the manuscript is improved.

No other changes are required.

Reviewer #2: The authors have done a good job of addressing the comments and the paper is stronger with these revisions. I would still recommend that a table of all the factors examined in the univariate regression was included, at least in the supplementary material, as I am sure that other readers will also wonder about the effect of age and gender.

Reviewer #3: The authors have responded to the comments and questions raised by the reviewers in a fully adequate fashion.

7. PLOS authors have the option to publish the peer review history of their article (what does this mean?). If published, this will include your full peer review and any attached files.

Reviewer #1: No

Reviewer #2: No

Reviewer #3: No

---

## [Editor Report · Acceptance letter]

31 Mar 2020

PONE-D-19-31902R1 

Prognostic impact of misdiagnosis of cardiac channelopathies as epilepsy 

Dear Dr. Jiménez-Jáimez:

I am pleased to inform you that your manuscript has been deemed suitable for publication in PLOS ONE. Congratulations! Your manuscript is now with our production department. 

With kind regards,

on behalf of

Dr Katriina Aalto-Setala 

Academic Editor

PLOS ONE